# Biochemical Pathways Leading to the Formation of Wyosine Derivatives in tRNA of Archaea

**DOI:** 10.3390/biom10121627

**Published:** 2020-12-02

**Authors:** Jaunius Urbonavičius, Daiva Tauraitė

**Affiliations:** Department of Chemistry and Bioengineering, Vilnius Gediminas Technical University, 10223 Vilnius, Lithuania; Daiva.Tauraite@vgtu.lt

**Keywords:** wyosine derivatives, tRNA modification, archaea, Trm5 enzymes, Taw enzymes

## Abstract

Tricyclic wyosine derivatives are present at position 37 in tRNA^Phe^ of both eukaryotes and archaea. In eukaryotes, five different enzymes are needed to form a final product, wybutosine (yW). In archaea, 4-demethylwyosine (imG-14) is an intermediate for the formation of three different wyosine derivatives, yW-72, imG, and mimG. In this review, current knowledge regarding the archaeal enzymes involved in this process and their reaction mechanisms are summarized. The experiments aimed to elucidate missing steps in biosynthesis pathways leading to the formation of wyosine derivatives are suggested. In addition, the chemical synthesis pathways of archaeal wyosine nucleosides are discussed, and the scheme for the formation of yW-86 and yW-72 is proposed. Recent data demonstrating that wyosine derivatives are present in the other tRNA species than those specific for phenylalanine are discussed.

## 1. Introduction

All transfer RNAs contain posttranscriptional modifications that are essential for the maintenance of the structure and proper functioning. Some of these modifications are chemically simple, whereas the other ones are complex, requiring several steps for biochemical synthesis [1]. One group of such modifications are the wyosine derivatives, tricyclic imidazopurines derived from the guanine, and found at position 37 of tRNA^Phe^ isolated from eukaryotes and Archaea [2]. The first report on such derivatives was published by RajBhandary and Chang more than 50 years ago, where the presence of a fluorescent derivative called Y-base was described in yeast *Saccharomyces cerevisiae* [3]. Wyosine derivatives were shown to be involved in codon–anticodon interactions [4], thereby preventing ribosomal -1 frameshifting [5,6].

The metabolic pathway leading to wybutosine (yW) in the cytoplasmic tRNA^Phe^ has been established first in yeast *Saccharomyces cerevisiae* using comparative genomics, knockout, and complementation [7]. This pathway includes five sequential reactions, each catalyzed by a distinct enzyme and starts with the formation of the N^1^-methylguanosine (m^1^G). Only the first reaction occurs in the nucleus [8]. In the fungi, *Torulopsis utilis*, wyosine (imG) is formed instead of wybutosine [9]. Similarly, the imG is found in mitochondria of a protist *Trypanosoma brucei* [10]. In contrast, several eukaryotes (*Drosophila melanogaster*, *Bombyx mori*, *Tinca tinca*) possess only m^1^G since no genes of the wyosine biosynthesis pathway are found in these organisms [11,12,13].

Several reports provide links between the synthesis of wyosine derivatives in human cells and cancer. It was observed that tRNA^Phe^ isolated from rat and mouse tumors did not contain a fully modified yW base that was present in normal tissues [14]. In addition, it was demonstrated that the human TYW2 gene is overexpressed in cancer [15]. Later, it was established that the function of hTYW2 is similar to the one of yeast counterpart [16] and that the loss of this protein induces -1 frameshifting in colon cancer [17].

In Archaea, several different wyosine derivatives are found [18]. Still, in euryarchaeon *Haloferax volcanii*, no wyosine derivatives, but only m^1^G is present in tRNA^Phe^ [19]. In many other Euryarchaeota, 7-aminocarboxypropyl-4-demethylwyosine (yW-86) and 7-aminocarboxypropyl-wyosine (yW-72) that were previously considered to be the intermediates of the eukaryotic pathway are found ([18,20], Scheme 1). For simplicity, trivial names are used when describing the wyosine derivatives. Compared to eukaryotes, two additional wyosine derivatives have been identified in Archaea: isowyosine (imG2) and 7-methylwyosine (mimG) ([20,21], Scheme 1). Moreover, in several archaeal species, a mixture of different wyosine derivatives has often been observed [18]. Recently, it was reported that five additional tRNAs isolated from *Methanocaldococcus jannaschii* contain either 4-demethylwyosine (imG-14) or wyosine (imG) at position 37 [22]. Moreover, the sequencing of tRNA^Trp^ isolated from *Thermococcus kodakarensis* demonstrated the presence of 7-methylwyosine at position 37 [23]. These surprising findings await further confirmation and the corresponding modification enzymes—biochemical and structural characterization.

## 2. Enzymatic Formation of 1-Methylguanosine, Intermediate in the Pathway Leading to the Formation of Wyosine Derivatives

Biosynthesis of wyosine derivatives in Archaea starts with methylation of guanosine by methyltransferases belonging to the Trm5 family of enzymes to produce 1-methylguanosine (m^1^G) that is present not only in tRNA^Phe^ but also in several other tRNA species (Scheme 1, see also [1]). The absence of m^1^G in different tRNA species induces the +1 translational frameshifting [24,25]. Interestingly, the formation of m^1^G in bacteria is catalyzed by the TrmD family of enzymes that are evolutionarily distinct from the Trm5 one, which in addition to Archaea, is also present in eukaryotes. These two families of enzymes have different structures and use different approaches for substrate recognition. TrmD and Trm5 enzymes have been compared in a recent review by Goto-Ito et al. [26]. TrmD family will not be further discussed here; instead, the focus will be made on the Trm5 family of Archaea.

Previously, it was assumed that Trm5 enzymes in Archaea consist of three subfamilies, Trm5a/b/c [18]. Our recent theoretical and experimental investigations [2,27] demonstrate that the Trm5a subfamily of enzymes belongs to wyosine biosynthesis pathways even though the Trm5a = Taw22 also catalyzes the formation of m^1^G (Scheme 1). Trm5a subfamily will be described below in the section that concerns the biosynthesis of isowyosine (imG2). Trm5b subfamily of enzymes is found in the Euryarchaeota group of Archaea, whereas Trm5c exists in Sulfolobales and Desulfurolcoccales orders of Crenoarchaeota [2]. A small conserved D1 domain is always present in both Trm5b and Trm5c, whereas it is absent in a majority of aTrm5a [18], which is a major structural difference between Trm5b/c and Trm5a; however, the differences between Trm5b and Trm5c families of enzymes are not that evident. It was described by J.Bujnicki that Trm5 proteins have a Rossman fold at C-termini for catalytic function, and also the NPPY motif at the fourth β-strand for positioning of N-atom of G37 substrate [28]. The crystal structures of Trm5b proteins were first determined for *Methanococcus jannaschii* (MjTrm5b) either without (Protein Data Bank (PDB) 2YX1) or with tRNA substrates (PDBs 2ZZM and 2ZZN) [29,30] and then for *Pyrococcus abyssi* (PaTrm5b) apo-enzyme (PDB 5YAC) [31]. In the latter work, it was demonstrated that the truncation of the D1 domain of PaTrm5b leads both to reduced binding to tRNA and enzymatic activity. To our knowledge, no crystal structures of Trm5c are available. The available structures of Trm5a enzymes and thereby structure-function relationships will be discussed in a section that concerns the formation of imG2 wyosine derivative.

## 3. Enzymatic Biosynthesis of Wyosine Derivatives

The other biosynthesis steps are specific for wyosine derivatives. The enzymes that catalyze such reactions in Archaea are named Taw for tRNA archaeal wyosine derivatives. Although these enzymes are often also named Tyw in the literature, it was suggested that such nomenclature should be used only for the yeast (or other eukaryotes), but not for archaeal enzymes, to make a distinction between the proteins present in the corresponding domain of life [18]. In this section, the biosynthesis of different intermediates and final products is described.

### 3.1. Formation of imG-14 by Taw1 Enzyme

Biosynthesis of wyosine derivatives in Archaea starts with the formation of the imidazo-purine ring yielding the 4-demethylwyosine (imG-14) that is catalyzed by the Taw1 enzyme. For the formation of different wyosine derivatives, *S*-adenosine-l-methionine (SAM) is used as a donor of the methyl or 3-amino-3-carboxypropyl (“acp”) groups. However, Taw1 is unique since it is a member of the radical SAM superfamily, which was identified based on a conserved CxxxCxxC motif [32] that is used for binding of [4Fe–4S] cluster. Here, SAM is used to activate the substrate rather than the chemical group donor. In Tyw1 of *S. cerevisiae*, a flavodoxin-like domain that binds an FMN cofactor is found [7]. In that work, it was demonstrated that conservative cysteines that bind [4Fe–4S] cluster are required for the formation of imG-14. To our knowledge, no data on structural and biochemical characterization of this or other eukaryotic proteins is available except for the unpublished structure of putative *Schizosaccharomyces japonicus* Tyw1 flavodoxin-like domain (6PUQ and 6PUP in Protein Data Bank); further characterization of this family of enzymes was done in Archaea. It was demonstrated that the formation of the imidazole ring from m^1^G requires two new carbon atoms, originating from the pyruvate, whose C2 and C3 atoms are incorporated into the imidazole ring [33,34]. Please note that the formation of the third ring changes the way the atoms are numbered (Scheme 1). Further structural, mechanistic, and spectroscopic studies on Taw1 proteins isolated either from *Pyrococcus horikoschii*, *P.abissi*, or *M.jannaschii*, revealed that they contain two [4Fe–4S] clusters [35,36,37,38] (Figure 1), and allowed to propose an enzymatic mechanism of the 3rd ring formation reaction, where the reductively cleaved SAM provides dAdo to initiate catalysis by abstracting a hydrogen atom from m^1^G, whereas second (so-called auxiliary) cluster is used for the activation of pyruvate by forming a Schiff base and decarboxylation [39,40], summarized in Scheme 2. More information about Taw1 and the other radical SAM enzymes may be found in a recent volume of Methods in Enzymology issue “Radical SAM enzymes” [41].

### 3.2. Formation of yW-86 by the Taw2 Enzyme

4-Demethylwyosine (imG-14) serves as a core compound for further modifications leading to the formation of either 7-aminocarboxypropyl-demethylwyosine (yW-86), wyosine (imG), or isowyosine (imG2). Taw2 enzyme belongs to the same cluster of orthologous groups (COG2520/PACE 25) as Trm5a/b/c methyltransferases [42,43]. These paralogs use the same SAM cofactor but transfer either the methyl group onto the guanosine or 3-amino-3-carboxypropyl (“acp”) groups onto the imG-14. The acp group containing tRNA modifications are known for bacteria and eukaryotes (acp^3^U20 and acp^3^U47); recently, the gene responsible for the formation of acp^3^U47 was identified in *Escherichia coli* [44,45]. Formation of yW-86 from imG-14 was demonstrated in vitro for the recombinant *M. jannaschii* and *P. horikoshii* Taw2 enzymes using tRNA isolated from yeast *tyw2* mutant and SAM cofactor [46]. In the same work, determination of the crystal structure (Figure 2A) and mutagenesis of the aforementioned enzymes revealed that juxtaposition and orientation of imG-14 (including the critical residue G199) and the absence of D1 domain are likely responsible for the transfer of the “acp”, but not the methyl group at the C7 position of the target base.

### 3.3. Formation of imG2 by Taw21/Taw22 (aTrm5) Enzymes

Isowyosine (imG2), together with 7-methylwyosine (mimG), are archaea-unique wyosine derivatives that are present in Thermococalles and several Crenarchaeota ([2,18] and references therein). It has been suggested that imG2 is the likely intermediate of mimG biosynthesis ([2,47], Figure 1). The occurrence of imG2 in different Archaea strictly correlates with the presence of archaeal Trm5a ([2] and references therein). As described above, the Archaeal Trm5a family is characterized by the absence (in many cases) of the D1 domain that is present in the Trm5b/c family of methyltransferases, responsible for the conversion of guanosine into 1-methylguanosine [27]. In some Archaea, like Thermoproteales order of Crenarchaeota, and nanoarchaeon *Nanoarchaeum equitans*, Trm5a is the only enzyme of the Trm5 family present, suggesting that it catalyzes the formation of not only imG2 but also of m^1^G. Indeed, it was demonstrated that recombinant Trm5a of *N. equitans* catalyzes the formation of both m^1^G and imG2. Based on these findings, we proposed to rename such dual-specificity Trm5a proteins into Taw22 ([2,27] Scheme 1). In addition to the formation of m^1^G in tRNA^Phe^, this enzyme is probably responsible for the formation of such modification in other tRNA species of *N. equitans*. A different situation exists in some Crenarchaeota, like Desulforococcales and Sulfolobales, where Trm5c is present alongside Trm5a and likely corresponds for the formation of m^1^G [2], whereas Trm5a is responsible for the formation of imG2. Indeed, we have demonstrated that Trm5a of *Sulfolobus solfataricus* catalyzes the formation of imG2, but not of m^1^G [27]. Based on these results, it was proposed to rename the monofunctional Trm5a enzymes into Taw21.

The situation is even more complicated in the Thermococcales order of Euryarchaeota. Here, Trm5b is present alongside the Trm5a. We have demonstrated that the Trm5a enzyme of *P. abyssi* catalyzes the formation of both imG2 and m^1^G in tRNA^Phe,^ but not in tRNA^Asp^ [18,27]. We conclude that in this organism, Taw22 acts on tRNA^Phe^, whereas Trm5b is responsible for the methylation of G37 in other m^1^G-containing tRNAs.

Several crystal structures of PaTrm5a=Taw22 with different cofactors and/or tRNA substrates were obtained (PDB 5HJJ, 5HJI, 5HJK, 5HJM, 5WT1, 5WT3) [48,49]. Structures of PaTrm5a without the tRNA substrate demonstrated an extended conformation (“open” state) compared to the well-folded MjTrm5b. Fluorescence resonance energy transfer (FRET) experiments demonstrated that such conformation is also a natural one in solution [48]. Subsequent SAXS (small-angle X-ray scattering) experiments confirmed the FRET data [49]. The co-crystal structures of PaTrm5a with tRNA^Phe^ substrate and S-adenosyl-L-homocysteine (SAH) or 5′-methylthioadenosine (MTA, a degradation product of SAM) let to propose the reaction mechanism of this enzyme. It was demonstrated that the orientation of the tRNA substrate and SAM cofactor towards either Trm5a or Trm5b is similar. The capability of Trm5a to catalyze the formation of imG2 in addition to m^1^G probably arises due to substrate recognition motif 260PTPK263 that is different from the conserved NPPY motif in other Trm5s. The side chain of Asn265 of MjTrm5b causes the steric clashes with the tricyclic ring produced by Taw1. In contrast, the PaTrm5a contains a proline instead of asparagine in the PTPK motif that allows the accommodation of the imG-14 substrate [49] (Figure 2B,C).

### 3.4. The Enzymatic Activity of Taw3 Proteins

The enzymes of the Taw3 family are the SAM-dependent methyltransferases that all belong to the COG1590/PACE24 group of proteins [42,43]. These enzymes catalyze three distinct reactions: formation of yW-72 from yW-86, imG from imG-14, and mimG from imG2. We propose to name such enzymatic activities Taw3a, Taw3b, and Taw3c, respectively (Scheme 1). In yeast, methylation of yW-86 to yield yW-72 is catalyzed by the Tyw3 enzyme. Orthologs of this enzyme are present in Archaea, and their enzymatic specificity must correlate with the presence of wyosine derivatives in different species. These relationships are discussed below and, for simplicity, are summarized in Table 1.

For example, two copies of Taw3 are found in *P. furiosus* and *P. abyssi* [18]. Such two enzymes must catalyze the formation of yW-72, imG, and mimG present in these organisms, thereby providing Taw3a/b/c activities. The preliminary results (Urbonavičius, unpublished) demonstrated that one of two Taw3 proteins from *P. abyssi* incorporates the radioactive methyl groups into tRNA isolated from *S. cerevisiae tyw2* (possessing the imG-14 substrate) and *tyw3* (possessing the yW-86 substrate) mutants, and *M. acetovorans* (possessing imG-14, imG2, and yW-86 substrates), suggesting that it has at least Taw3a/b and possibly Taw3a/b/c activities. In contrast, another Taw3 enzyme incorporates the radioactive methyl groups only into tRNA isolated from *S.cerevisiae tyw2* mutant (possessing the imG-14 substrate, Taw3b activity). However, more experiments are needed to confirm these results.

For several Archaea, it is evident that Taw3 enzymes methylate imG-14 to form imG (Taw3b activity) since they contain only one copy of the corresponding gene in their genomes, and neither yW-72 nor imG2 is found in these organisms. This applies to several Methanococcales such as *M. jannaschii* and *Methanothermococcus litotrophus* [2]. The enzymatic activity of these enzymes may be elucidated by incubating them with tRNA isolated from yeast *tyw2* mutant (possessing imG-14) and monitoring the formation of the imG-14 product by HPLC/MS or TLC as performed previously for the other Tyw/Taw enzymes [7,18,27].

In several Crenarchaeota like *Sulfolobus solfataricus*, also in Thermoproteales *Thermoproteus neutropylus*, *Pyrobaculum calidifontis*, and *Pyrobaculum islandicum*, single Taw3 enzymes likely methylate imG2 into mimG because these two wyosine derivatives (and sometimes the imG-14 intermediate) are found in tRNA, whereas imG and yW-72 are not [2]. We propose to name these enzymes Taw3c. In recent work, the crystal structure of putativeTaw3c protein from *S. solfataricus* was solved and revealed a novel α/β fold, which constitutes a new class of methyltransferases composed of SHS2, RAGNYA fold domains, and an N-terminal extension (PDB 1TLJ, [50], Figure 3).

Interestingly, total tRNA isolated from *S. cerevisiae tyw3* mutant and possessing yW-86 was not a substrate for SsTaw3c enzyme, clearly distinguishing it from yeast Tyw3 or *Pyrococcus* Taw3a proteins. Transfer RNA isolated from *S. cerevisiae tyw2* mutant and possessing imG-14 was also not a substrate for *S. solfataricus* Taw3c, clearly distinguishing it from Taw3b. No homologous tRNA substrate isolated from *S. solfataricus tyw3* mutant (possessing imG2) was used. Still, the development of the *Sulfolobus* genetics tools now allows the creation of such mutants in three closely related archaea, *S. sulfotaricus*, *S. acidocaldarius*, and *S. islandicus* [51,52,53], which would allow establishing the enzymatic activity of Taw3c.

In the aforementioned work [50], the structure of SsTaw3c was compared with the other three Taw3 structures deposited into PDB: *Archaeoglobus fulgidus* (PDB 2QG3), *Pyrococcus horikoshii* (PDB 2IT2), and *Aeropyrum pernix* (PDB 2DVK)). It was shown that all three structures share the same domain architecture and overall fold as SsTaw3. However, it is not clear whether these Taw3 enzymes belong to the Taw3c subgroup. Two copies of Taw3 enzymes are present in *P. horikoshii* [18], similarly as in *P. abyssi* and *P. furiosus*; the enzymatic activity of the copy with solved structure (PH1069) is unknown. In *A. fulgidus*, Trm5a, Taw1, Taw2, and one copy of Taw3 is present [18], but no mimG is found in tRNA [54]. Therefore, it is possible that AF2059 (AfTaw3) protein is responsible for the formation of either yW-72 (Taw3a activity) or imG (Taw3b activity), or both. Only APE0816 (ApTaw3) protein is likely to have a Taw3c activity, based on the similar distribution of Taw proteins and phylogenetic closeness to *Sulfolobus* [18].

Interestingly, *Pyrolobus fumarii* contains two copies of Taw3, and both imG and mimG [2]. Therefore, it is likely that one of these copies is responsible for the formation of imG (Taw3b activity), whereas another one—for the formation of mimG (Taw3c activity). In *Steteria hydrogenophyla* and *Staphylothermus marinus*, a single copy of Taw3 seems to be responsible for the formation of both imG and mimG, therefore possessing Taw3b/c activities [2,18,55]. However, these predictions await experimental confirmation.

## 4. Chemical Synthesis of Wyosine Nucleosides

Determination of the identity of naturally occurring compounds requires their isolation and comparison of structure with corresponding chemically synthesized derivatives. Usually, the wyosine derivatives are analyzed by isolation of tRNA, digestion of it to the nucleotides and then nucleosides, and performing the HPLC/MS analysis [7,27]. In this part, the chemical synthesis of wyosine nucleosides that are present in Archaea will be described. The rationale for the chemical synthesis of nucleosides differs from the biochemical one. Synthesis schemes are similar for the nucleosides that differ only by the presence of one methyl group at certain positions; therefore, they will be described pairwise.

### 4.1. Synthesis of imG-14 and imG2

Synthesis of both compounds is performed by the formation of imidazo-ring from guanosine to form N1-N^2^ two-carbon bridge; the first report of imG-14 synthesis being published by Kasai and coworkers in 1976 [21,27,56,57]. To obtain demethylwyosine (imG-14), guanosine is reacted with chloro- or bromoacetone after treatment of guanosine with sodium hydride (Scheme 3).

Similarly, isowyosine (imG2) is obtained from guanosine using either 3-bromo-2-butanone or 3-chloro-2-butanone. This one-step reaction does not require the protection of hydroxy groups of the guanosine and leads to the strongly fluorescent tricyclic nucleosides at the yield of 40–60%.

### 4.2. Synthesis of imG and mimG

The wyosine (imG) and methylwyosine (mimG) may be obtained similarly as imG-14 and imG2 by ring closure reaction, starting from 3-methylguanosine (m^3^G) [58,59]. However, 3-methylguanosine is not commercially available and cannot be obtained by direct methylation of the guanosine. The synthesis of m^3^G is a laborious multistep procedure that requires the usage of some harmful reagents, such as cyanogen bromide, Vielsmeier reagent (CH_3_)_2_ N^+^=CHCl^−^, long reaction times (e.g., 240 h methylation with methyl iodide), and also repeating the protection/deprotection steps of hydroxy groups [60,61]. Scheme 4 depicts the 10-step synthesis of imG or mimG starting from the inosine (I) via key intermediate 3-methylguanosine, following by ring closure reaction with overall yield only up to 1% (Tauraitė, unpublished).

The alternative synthesis route for the imG and mimG starts from the tricyclic imG or mimG compounds by the direct methylation of N4-position with diethylzinc ZnEt_2_ and iodomethane CH_2_I_2_ [62,63]. In contrast to guanine, the tricyclic heterocyclic base can be methylated directly and regioselectively at N4-position using an organozinc reagent, produced in situ by the reaction of iodomethane with diethylzinc in dimethoxyethane (glyme). This synthesis route is shorter, more efficient, and leads to an overall yield of 10–15%. It starts from the guanosine that is first converted to imG-14 or imG2 derivatives (Scheme 3) that are subsequently methylated, producing imG or mimG (Scheme 5).

### 4.3. Synthesis of yW-86 and yW-72

As was mentioned above, chemically synthesized wyosine derivatives are necessary for the determination of the identity of modified nucleosides isolated from tRNA. To our knowledge, 7-aminocarboxypropyl-demethylwyosine (yW-86) and 7-amino-carboxypropyl-wyosine (yW-72) have never been chemically synthesized. Their presence in tRNA is inferred from the HPLC/MS analysis data without comparison with chemically synthesized derivatives.

We propose a synthesis scheme that is based on the previous work of Itaya and Carell groups [63,64,65]. Synthesis of yW-86 or yW-72 starts from either imG-14 or imG. One of the possible synthesis routes is via an iodinated key intermediate, which is obtained by the introduction of iodine at position C7 of either compound. The next step is the Heck coupling between 7-iodo-tricyclic nucleosides and the vinyl glycine. Following catalytic hydrogenation with palladium on carbon and the deprotection of the hydroxy groups, the desired hypermodified wyosine derivatives yW-86 and yW-72 may be obtained (Scheme 6).

## 5. Conclusions and Future Directions

In this review, we summarized the current knowledge regarding one of the most complex biosynthetic pathways in archaeal tRNA—biosynthesis of the wyosine derivatives. Recent biochemical and structural work elucidated the enzymatic mechanisms leading to the formation of the imidazopurine ring from 1-methylguanosine catalyzed by Taw1 enzyme that uses 5′-deoxyadenosine radical to initiate the 3rd ring cloture. It was also established that further modifications are catalyzed by distinct enzymes belong to the COG2520 family (the same that participates in the methylation of guanosine) that use the SAM cofactor to provide either the methyl- or 3-amino-3-carboxypropyl groups to further modify the imidazopurine ring. Whereas Taw2 enzymes catalyze the formation of yW-86, either the monofunctional (Trm5a = Taw21) or bifunctional (Trm5a = Taw22) enzymes add the methyl group onto imG-14 to form the imG2. Structural and biochemical characterization of Taw21 proteins is needed to understand why their enzymatic activity differs from those of Taw22 and Trm5b/c.

We also noted that the knowledge regarding the last steps of the formation of the wyosine derivatives in Archaea is scarce since no enzymatic activity of the Taw3 family of proteins that are homologs of *S. cerevisiae* Tyw3 and belong to the COG1590 family was demonstrated. We proposed that this family of enzymes possess at least three distinct enzymatic activities. To elucidate such enzymatic activities, different wyosine nucleosides are required. Two of those, yW-86 and yW-72, have not been chemically synthesized previously. Here, we propose the scheme for the total synthesis of such nucleosides. By this, we provide the synthesis routes for all archaea-specific wyosine derivatives.

The recent development of direct tRNA sequencing methods led to the determination of wyosine derivatives in several tRNA species that are not specific for phenylalanine. Therefore, the currently accepted paradigm that wyosine derivatives are present only in tRNA^Phe^ may be reconsidered in the near future, and previously obtained data regarding the activities of the enzymes involved in the formation of wyosine derivatives may be revised.

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
