# Peer review of "Biochemical Pathways Leading to the Formation of Wyosine Derivatives in tRNA of Archaea"

_biomolecules, 2020, doi:10.3390/biom10121627_

Round 1
Reviewer 1 Report
The manuscript by Urbonavicius and Tauraite is a review of the biosynthesis of the tRNA modification wyosine and its derivatives, and while the focus is on the archaeal pathways, the authors also discuss relevant related systems in Bacteria (in the case of m1G) and Eukarya when appropriate. The authors provide a brief summary of background material related to biological function to provide some context to the review, and do a good job of covering the recent literature – approximately 1/3 of the references are from 2014 or sooner, the time of the last review. However, only one figure in the manuscript is devoted to covering the pathways (see below for more on this), and no figures are provided that illustrate the biochemical mechanisms of the enzymatic steps (e.g. for Taw1) or illustrate the interesting structural biology of the wyosine pathways (e.g. as illustrated with the Taw22 enzymes).
Curiously, while the manuscript is titled “Biochemical pathways leading to the formation of wyosine derivatives in Archaea,” a substantial portion of it (Section 4), including 4 of the 5 figures, is dedicated instead to the chemical synthesis of these molecules. Furthermore, section 4.3 (and Figure 5) addresses not published work, but a proposed synthesis of yW-86 and yW-72. It is certainly common to see proposals of possible steps and mechanisms in biochemical pathways presented in the primary and review literature, but I think that this is the first time I’ve ever seen a proposed chemical synthesis complete with suggested reagents. I was left wondering what purpose its inclusion was intended to serve. While I certainly agree with the authors that the identification of modified nucleosides present in tRNA is ultimately dependent on standards of known composition prepared through chemical synthesis, this section largely presents work done decades ago - only one paper is cited that was published since 2014 – and doesn’t support the focus on new biochemical discoveries elsewhere in the manuscript. If the authors wish to cover both biological and chemical systems the title of the manuscript should be changed (but section 4.3 should still be cut) and the overall approach to the manuscript changed from a focus on recent advances to a comprehensive review.
The following changes are needed before publication:
- The manuscript needs a thorough edit for grammar and spelling. For example, there are many examples of the misuse of the articles “the” and “a” throughout the manuscript – i.e. including them where they don’t belong and excluding them where they do. The phrases “By contrary” and “In contrary” are used several times and should be replaced with “In contrast.”
- If the authors keep the sections on chemical synthesis, the figure files should all be checked as there are many formatting problems in my review copy.
Author Response
Point 1: no figures are provided that illustrate the biochemical mechanisms of the enzymatic steps (e.g. for Taw1).
Response 1: Both crystal structures (Figure 1) and Scheme1 that illustrate the biochemical mechanism of Taw1 are provided.
Point 2: Curiously, while the manuscript is titled “Biochemical pathways leading to the formation of wyosine derivatives in Archaea,” a substantial portion of it (Section 4), including 4 of the 5 figures, is dedicated instead to the chemical synthesis of these molecules. Furthermore, section 4.3 (and Figure 5) addresses not published work, but a proposed synthesis of yW-86 and yW-72. It is certainly common to see proposals of possible steps and mechanisms in biochemical pathways presented in the primary and review literature, but I think that this is the first time I’ve ever seen a proposed chemical synthesis complete with suggested reagents. I was left wondering what purpose its inclusion was intended to serve. While I certainly agree with the authors that the identification of modified nucleosides present in tRNA is ultimately dependent on standards of known composition prepared through chemical synthesis, this section largely presents work done decades ago - only one paper is cited that was published since 2014 – and doesn’t support the focus on new biochemical discoveries elsewhere in the manuscript. If the authors wish to cover both biological and chemical systems the title of the manuscript should be changed (but section 4.3 should still be cut) and the overall approach to the manuscript changed from a focus on recent advances to a comprehensive review.
Response 2: Since in the Scheme 1 the biochemical pathways leading to the formation of 6 wyosine derivatives are presented, we think that it is useful to have the chemical synthesis routes for all these derivatives summarized in one paper. To our knowledge, no such compilation was done before. We prefer to keep the section 4.3 and simplified synthesis scheme for yW-86 and yW-72 since this is essential for the determination of the identity of these wyosine derivatives in tRNA as stated by the Reviewer. We also would like to keep the title since the chemistry provided here serves only as a tool to confirm the biochemical data.
Point 3: The manuscript needs a thorough edit for grammar and spelling. For example, there are many examples of the misuse of the articles “the” and “a” throughout the manuscript – i.e. including them where they don’t belong and excluding them where they do. The phrases “By contrary” and “In contrary” are used several times and should be replaced with “In contrast.”
Response 3: The text was checked for spelling and the use of the articles. The phrases "By contrary" and "In contrary" were replaced by "In contrast".
Point 4: If the authors keep the sections on chemical synthesis, the figure files should all be checked as there are many formatting problems in my review copy.
Response 4: The figure files were checked again before the resubmission. If the problem persists, the authors are ready to provide the original figures for the evaluation/publication.
Reviewer 2 Report
The authors in the present manuscript offer the reader a sufficient overview of the biosynthesis of wyosine derivatives in Achaea and of the related enzymatic activities. The most important aspect, in my opinion, is the proposed re-classification of enzymes in this domain of life, appropriately clarifying the existing differences with the homologues present in the other domains. However, given the structural information present in the literature, I believe that this manuscript can be improved with some illustrations of the primary and (where available) 3D structures, to better describe some parts in the text. In general, this review is interesting for a specific audience on this topic and I recommend it for publication on MDPI Biomolecules.
Author Response
Point 1: However, given the structural information present in the literature, I believe that this manuscript can be improved with some illustrations of the primary and (where available) 3D structures, to better describe some parts in the text.
Response 1: We provided several 3D structures to illustrate the points described in the manuscript.
Reviewer 3 Report
This is a timely review on a complex topic: the formation of wyosine derivatives in Archaea.
These hypermodifications of tRNAs have important roles in decoding accuracy and one of the most complex pathways found for the synthesis of modified bases.
If the content of the manuscript is accurate and reviews well the current status of the field, the english is quite poor making the text inaccurate or confusing at time.
For example:
Line 74 the transition word “However “ is not appropriate as the fact that there are three Trm5 subfamilies does not preclude from them being involved in different pathways.
The whole section on Taw3 is nearly impossible to follow.
In addition to fixing the english, the review lacks figures to explain the Taw2 and Taw3 subfamilies. Figures showing the differences in structures and or motifs would make it much easier for the reader to follow. I think doing sequence similarity networks (SSNs) at least for the Taw3 family would also greatly reinforce the points made by the authors.
Author Response
Point 1: If the content of the manuscript is accurate and reviews well the current status of the field, the english is quite poor making the text inaccurate or confusing at time.
Point 1: The text was proof-read to correct for misspellings and to provide the correct usage of the articles.
Point 2: Line 74 the transition word “However “ is not appropriate as the fact that there are three Trm5 subfamilies does not preclude from them being involved in different pathways.
Response 2: This sentence was corrected and now stands as follows: "Our recent theoretical and experimental investigations [2,27] demonstrate that the Trm5a subfamily of enzymes belongs to wyosine biosynthesis pathways even though the Trm5a=Taw22 also catalyzes the formation of m1G (Scheme 1)."
Point 3: The whole section on Taw3 is nearly impossible to follow.
Response 3: This section was split into several short sections for clarity. Figure 3 and Table 1 were added to better explain our suggestions regarding the enzymatic activity of Taw3 enzymes.
Point 4. Figures showing the differences in structures and or motifs would make it much easier for the reader to follow.
Response 4: Three additional figures depicting the crystal structures of Taw enzymes with the important features highlighted, a scheme describing the proposed enzymatic mechanism of Taw1, and a table with proposed the enzymatic activities of Taw3 enzymes were added to help the readers to follow the text.
Point 5. I think doing sequence similarity networks (SSNs) at least for the Taw3 family would also greatly reinforce the points made by the authors.
Response 5: Many thanks for the pointing at interesting bioinformatics method that we haven't used before. We have performed SSN analysis following the instructions provided by Gerlt et. al. in Biochimica et Biophysica Acta, 2015. Indeed, using the "default" setup in "option A" suggested by the authors, we obtained at least 1 large and 3 small subfamilies of Taw3 proteins. Still, since this method requires the manual adjustments that have to be precisely described, we believe that this kind of analysis is more suitable for the experimental/bioinformatics article rather than for a review paper. We prefer to add Table 1 to point directly at those Taw3 proteins whose enzymatic activities could be tested experimentally.
This manuscript is a resubmission of an earlier submission. The following is a list of the peer review reports and author responses from that submission.